# Deep active inference agents using Monte-Carlo methods

**Zafeirios Fountas**[*]
Emotech Labs &
WCHN, University College London
`f@emotech.co`

**Noor Sajid**
WCHN, University College London
`noor.sajid.18@ucl.ac.uk`

**Pedro A.M. Mediano**
University of Cambridge
`pam83@cam.ac.uk`

**Karl Friston**
WCHN, University College London
`k.friston@ucl.ac.uk`

## Abstract

Active inference is a Bayesian framework for understanding biological intelligence. The underlying theory brings together perception and action under one single imperative: minimizing free energy. However, despite its theoretical utility in explaining intelligence, computational implementations have been restricted to low-dimensional and idealized situations. In this paper, we present a neural architecture for building deep active inference agents operating in complex, continuous state-spaces using multiple forms of Monte-Carlo (MC) sampling. For this, we introduce a number of techniques, novel to active inference. These include: $i)$ selecting free-energy-optimal policies via MC tree search, $ii)$ approximating this optimal policy distribution via a feed-forward 'habitual' network, $iii)$ predicting future parameter belief updates using MC dropouts and, finally, $iv)$ optimizing state transition precision (a high-end form of attention). Our approach enables agents to learn environmental dynamics efficiently, while maintaining task performance, in relation to reward-based counterparts. We illustrate this in a new toy environment, based on the dSprites data-set, and demonstrate that active inference agents automatically create disentangled representations that are apt for modeling state transitions. In a more complex Animal-AI environment, our agents (using the same neural architecture) are able to simulate future state transitions and actions (i.e., plan), to evince reward-directed navigation - despite temporary suspension of visual input. These results show that deep active inference – equipped with MC methods – provides a flexible framework to develop biologically-inspired intelligent agents, with applications in both machine learning and cognitive science.

## 1 Introduction

A common goal in cognitive science and artificial intelligence is to emulate biological intelligence, to gain new insights into the brain and build more capable machines. A widely-studied neuroscience proposition for this is the free-energy principle, which views the brain as a device performing variational (Bayesian) inference [1, 2]. Specifically, this principle provides a framework for understanding biological intelligence, termed active inference, by bringing together perception and action under a single objective: minimizing free energy across time [3–7]. However, despite the potential of active inference for modeling intelligent behavior, computational implementations have been largely restricted to low-dimensional, discrete state-space tasks [8–11].

---

[*]Corresponding author

Recent advances have seen deep active inference agents solve more complex, continuous state-space tasks, including Doom [12], the mountain car problem [13–15], and several tasks based on the MuJoCo environment [16], many of which use amortization to scale-up active inference [13–15, 17]. A common limitation of these applications is a deviation from vanilla active inference in their ability to plan. For instance, Millidge [17] introduced an approximation of the agent's *expected free energy* (EFE), the quantity that drives action selection, based on bootstrap samples, while Tschantz *et al.* [16] employed a reduced version of EFE. Additionally, since all current approaches tackle low-dimensional problems, it is unclear how they would scale up to more complex domains. Here, we propose an extension of previous formulations that is closely aligned with active inference [4, 9] by estimating all EFE summands using a single deep neural architecture.

Our implementation of deep active inference focuses on ensuring both scalability and biological plausibility. We accomplish this by introducing Monte-Carlo (MC) sampling – at several levels – into active inference. For planning, we propose the use of MC tree search (MCTS) for selecting a free-energy-optimal policy. This is consistent with planning strategies employed by biological agents and provides an efficient way to select actions (see Sec. 5). Next, we approximate the optimal policy distribution using a feed-forward 'habitual' network. This is inspired by biological habit formation, when acting in familiar environments that relieves the computational burden of planning in commonly-encountered situations. Additionally, for both biological consistency and reducing computational burden, we predict model parameter belief updates using MC-dropouts, a problem previously tackled with networks ensembles [16]. Lastly, inspired by neuromodulatory mechanisms in biological agents, we introduce a top-down mechanism that modulates precision over state transitions, which enhances learning of latent representations.

In what follows, we briefly review active inference. This is followed by a description of our deep active inference agent. We then evaluate the performance of this agent. Finally, we discuss the potential implications of this work.

## 2  Active Inference

Agents defined under active inference: $A$) sample their environment and calibrate their internal generative model to best explain sensory observations (i.e., reduce surprise) and $B$) perform actions under the objective of reducing their uncertainty about the environment. A more formal definition requires a set of random variables: $s_{1:t}$ to represent the sequence of hidden states of the world till time $t$, $o_{1:t}$ as the corresponding observations, $\pi = \{a_1, a_2, ..., a_T\}$ as a sequence of actions (typically referred to as 'policy' in the active inference literature) up to a given time horizon $T \in \mathbb{N}^+$, and $P_\theta(o_{1:t}, s_{1:t}, a_{1:t-1})$ as the agent's generative model parameterized by $\theta$ till time $t$. From this, the agent's surprise at time $t$ can be defined as the negative log-likelihood $-\log P_\theta(o_t)$. Through slight abuse of notation, $P_\theta(.)$ denotes distribution parameterisation by $\theta$ and $P(\theta)$ denotes use of that particular distribution as a random variable. See supplementary materials for definitions (Table 1).

To address objective $A$) under this formulation, the surprise of current observations can be indirectly minimized by optimizing the parameters, $\theta$, using as a loss function the tractable expression:

$$-\log P_\theta(o_t) \leq \mathbb{E}_{Q_\phi(s_t, a_t)} \big[ \log Q_\phi(s_t, a_t) - \log P_\theta(o_t, s_t, a_t) \big] , \qquad (1)$$

where $Q_\phi(s_t, a_t)$ is an arbitrary distribution of $s_t$ and $a_t$ parameterized by $\phi$. The RHS expression of this inequality is the *variational free energy* at time $t$. This quantity is commonly referred to as negative *evidence lower bound* [18] in variational inference. Furthermore, to realize objective $B$), the expected surprise of future observations $-\log P(o_\tau|\theta, \pi)$ where $\tau \geq t$ – can be minimized by selecting the policy that is associated with the lowest EFE, $G$ [19]:

$$G(\pi, \tau) = \mathbb{E}_{P(o_\tau|s_\tau, \theta)} \mathbb{E}_{Q_\phi(s_\tau, \theta|\pi)} \big[ \log Q_\phi(s_\tau, \theta|\pi) - \log P(o_\tau, s_\tau, \theta|\pi) \big] , \qquad (2)$$

Finally, the process of action selection in active inference is realized as sampling from the distribution

$$P(\pi) = \sigma\big( -G(\pi) \big) = \sigma\Big( -\sum_{\tau > t} G(\pi, \tau) \Big) , \qquad (3)$$

where $\sigma(\cdot)$ is the softmax function.

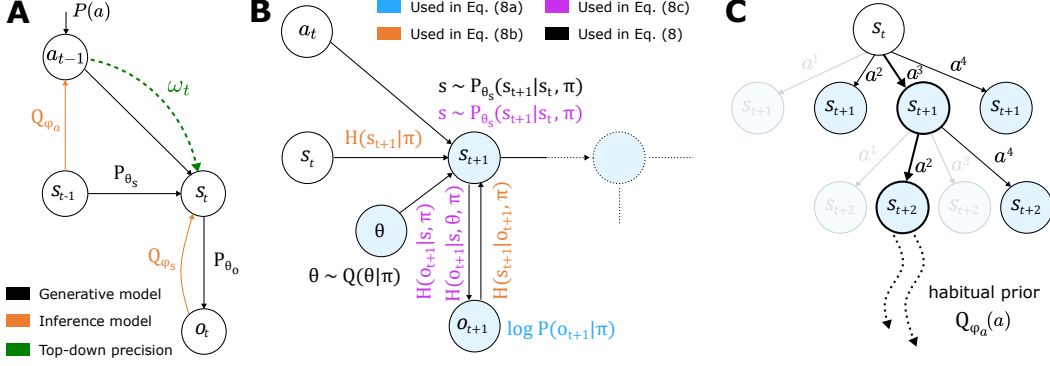

Figure 1: **A**: Schematic of model architecture and networks used during the learning process. Black arrows represent the generative model ($P$), orange arrows the recognition model ($Q$), and the green arrow the top-down attention ($\omega_t$). **B**: Relevant quantities for the calculation of EFE $G$, computed by simulating the future using the generative model and ancestral sampling. Where appropriate, expectations are taken with a single MC sample. **C**: MCTS scheme used for planning and acting, using the habitual network to selectively explore new tree branches.

## 3 Deep Active Inference Architecture

In this section, we introduce a deep active inference model using neural networks, based on amortization and MC sampling.

Throughout this section, we denote the parameters of the generative and recognition densities with $\theta$ and $\phi$, respectively. The parameters are partitioned as follows: $\theta = \{\theta_o, \theta_s\}$, where $\theta_o$ parameterizes the observation function $P_{\theta_o}(o_t|s_t)$, and $\theta_s$ parameterizes the transition function $P_{\theta_s}(s_\tau|s_t, a_t)$. For the recognition density, $\phi = \{\phi_s, \phi_a\}$, where $\phi_s$ is the amortization parameters of the approximate posterior $Q_{\phi_s}(s_t)$ (i.e., the state encoder), and $\phi_a$ the amortization parameters of the approximate posterior $Q_{\phi_a}(a_t)$ (i.e., our habitual network).

### 3.1 Calculating variational and expected free energy

First, we extend the probabilistic graphical model (as defined in Sec. 2) to include the action sequences $\pi$ and factorize the model based on Fig. 1A. We then exploit standard variational inference machinery to calculate the free energy for each time-step $t$ as:

$$F_t = - \mathbb{E}_{Q_{\phi_s}(s_t)} \left[ \log P_{\theta_o}(o_t|s_t) \right] + D_{\text{KL}} \left[ Q_{\phi_s}(s_t) \parallel P_{\theta_s}(s_t|s_{t-1}, a_{t-1}) \right]$$
$$+ \mathbb{E}_{Q_{\phi_s}(s_t)} \left[ D_{\text{KL}} \left[ Q_{\phi_a}(a_t) \parallel P(a_t) \right] \right] , \tag{4}$$

where

$$P(a) = \sum_{\pi:a_1=a} P(\pi) \tag{5}$$

is the summed probability of all policies that begin with action $a$. We assume that $s_t$ is normally distributed and $o_t$ is Bernoulli distributed, with all parameters given by a neural network, parameterized by $\theta_o$, $\theta_s$, and $\phi_s$ for the observation, transition, and encoder models, respectively (see Sec. 3.2 for details about $Q_{\phi_a}$). With this assumption, all the terms here are standard log-likelihood and KL terms easy to compute for Gaussian and Bernoulli distributions. The expectations over $Q_{\phi_s}(s_t)$ are taken via MC sampling, using a single sample from the encoder.

Next, we consider EFE. At time-step $t$ and for a time horizon up to time $T$, EFE is defined as [4]:

$$G(\pi) = \sum_{\tau=t}^{T} G(\pi, \tau) = \sum_{\tau=t}^{T} \mathbb{E}_{\tilde{Q}} \left[ \log Q(s_\tau, \theta|\pi) - \log \tilde{P}(o_\tau, s_\tau, \theta|\pi) \right] , \tag{6}$$

where $\tilde{Q} = Q(o_\tau, s_\tau, \theta|\pi) = Q(\theta|\pi)Q(s_\tau|\theta, \pi)Q(o_\tau|s_\tau, \theta, \pi)$ and $\tilde{P}(o_\tau, s_\tau, \theta|\pi) = P(o_\tau|\pi)Q(s_\tau|o_\tau)P(\theta|s_\tau, o_\tau)$. Following Schwartenbeck *et al.* [20], the EFE of a single time instance

$\tau$ can be further decomposed as

$$G(\pi, \tau) = - \mathbb{E}_{\tilde{Q}} \left[ \log P(o_\tau | \pi) \right] \tag{7a}$$

$$+ \mathbb{E}_{\tilde{Q}} \left[ \log Q(s_\tau | \pi) - \log P(s_\tau | o_\tau, \pi) \right] \tag{7b}$$

$$+ \mathbb{E}_{\tilde{Q}} \left[ \log Q(\theta | s_\tau, \pi) - \log P(\theta | s_\tau, o_\tau, \pi) \right]. \tag{7c}$$

Interestingly, each term constitutes a conceptually meaningful expression. The term (7a) corresponds to the likelihood assigned to the desired observations $o_\tau$, and plays an analogous role to the notion of reward in the reinforcement learning (RL) literature [21]. The term (7b) corresponds to the mutual information between the agent's beliefs about its latent representation of the world, before and after making a new observation, and hence, it reflects a motivation to explore areas of the environment that resolve state uncertainty. Similarly, the term (7c) describes the tendency of active inference agents to reduce their uncertainty about model parameters via new observations and is usually referred to in the literature as active learning [3], novelty, or curiosity [20].

However, two of the three terms that constitute EFE cannot be easily computed as written in Eq. (7). To make computation practical, we will re-arrange these expressions and make further use of MC sampling to render these expressions tractable and re-write Eq. (7) as

$$G(\pi, \tau) = - \mathbb{E}_{Q(\theta|\pi)Q(s_\tau|\theta,\pi)Q(o_\tau|s_\tau,\theta,\pi)} \left[ \log P(o_\tau | \pi) \right] \tag{8a}$$

$$+ \mathbb{E}_{Q(\theta|\pi)} \left[ \mathbb{E}_{Q(o_\tau|\theta,\pi)} H(s_\tau | o_\tau, \pi) - H(s_\tau | \pi) \right] \tag{8b}$$

$$+ \mathbb{E}_{Q(\theta|\pi)Q(s_\tau|\theta,\pi)} H(o_\tau | s_\tau, \theta, \pi) - \mathbb{E}_{Q(s_\tau|\pi)} H(o_\tau | s_\tau, \pi), \tag{8c}$$

where these expressions can be calculated from the deep neural network illustrated in Fig. 1B. The derivation of Eq. (8) can be found in the supplementary material. To calculate the terms (8a) and (8b), we sample $\theta$, $s_\tau$ and $o_\tau$ sequentially (through ancestral sampling) and then $o_\tau$ is compared with the prior distribution $\log P(o_\tau | \pi)$ The parameters of the neural network $\theta$ are sampled from $Q(\theta)$ using the MC dropout technique [22]. Similarly, to calculate the expectation of $H(o_\tau | s_\tau, \pi)$, the same drawn $\theta$ is used again and $s_\tau$ is re-sampled for $N$ times while, for $H(o_\tau | s_\tau, \theta, \pi)$, the set of parameters $\theta$ is also re-sampled $N$ times. Finally, all entropies can be computed using the standard formulas for multivariate Gaussian and Bernoulli distributions.

## 3.2 Action selection and the habitual network

In active inference, agents choose an action given by their EFE. In particular, any given action is selected with a probability proportional to the accumulated negative EFE of the corresponding policies $G(\pi)$ (see Eq. (3) and Ref. [19]). However, computing $G$ across all policies is costly since it involves making an exponentially-increasing number of predictions for $T$-steps into the future, and computing all the terms in Eq. (8). To solve this problem, we employ two methods operating in tandem. First, we employ standard MCTS [23–25], a search algorithm in which different potential future trajectories of states are explored in the form of a search tree (Fig. 1C), giving emphasis to the most likely future trajectories. This algorithm is used to calculate the distribution over actions $P(a_t)$, defined in Eq. (5), and control the agent's final decisions. Second, we make use of amortized inference through a habitual neural network that directly approximates the distribution over actions, which we parameterize by $\phi_a$ and denote $Q_{\phi_a}(a_t)$ – similarly to Refs. [26–28]. In essence, $Q_{\phi_a}(a_t)$ acts as a variational posterior that approximates $P(a_t | s_t)$, with a prior $P(a_t)$, calculated by MCTS (see Fig. 1A). During learning, this network is trained to reproduce the last executed action $a_{t-1}$ (selected by sampling $P(a_t)$) using the last state $s_{t-1}$. Since both tasks used in this paper (Sec. 4) have discrete action spaces $\mathcal{A}$, we define $Q_{\phi_a}(a_t)$ as a neural network with parameters $\phi_a$ and $|\mathcal{A}|$ softmax output units.

During the MCTS process, the agent generates a weighted search tree iteratively that is later sampled during action selection. In each single MCTS loop, one plausible state-action trajectory $(s_t, a_t, s_{t+1}, a_{t+1}, ..., s_\tau, a_\tau)$ – starting from the present time-step $t$ – is calculated. For states that are explored for the first time, the distribution $P_{\theta_s}(s_{t+1} | s_t, a_t)$ is used. States that have been explored are stored in the *buffer* search tree and accessed during later loops of the same planning process. The weights of the search tree $\tilde{G}(s_t, a_t)$ represent the agent's best estimation for EFE after taking action $a_t$ from state $s_t$. An upper confidence bound for $G(s_t, a_t)$ is defined as

$$U(s_t, a_t) = \tilde{G}(s_t, a_t) + c_{\text{explore}} \cdot Q_{\phi_a}(a_t | s_t) \cdot \frac{1}{1 + N(a_t, s_t)}, \tag{9}$$

where $N(a_t, s_t)$ is the number of times that $a_t$ was explored from state $s_t$, and $c_{\text{explore}}$ a hyper-parameter that controls exploration. In each round, the EFE of the newly-explored parts of the trajectory is calculated and back-propagated to all visited nodes of the search tree. Additionally, actions are sampled in two ways. Actions from states that have been explored are sampled from $\sigma(U(a_t, s_t))$ while actions from new states are sampled from $Q_{\phi_a}(a_t)$.

Finally, the actions that assemble the selected policy are drawn from $P(a_t) = \frac{N(a_t, s_t)}{\sum_j N(a_{j,t}, s_t)}$. In our implementation, the planning loop stops if either the process has identified a clear option (i.e. if $\max P(a_t) - 1/|\mathcal{A}| > T_{dec}$) or the maximum number of allowed loops has been reached.

Through the combination of the approximation $Q_{\phi_a}(a_t)$ and the MCTS, our agent has at its disposal two methods of action selection. We refer to $Q_{\phi_a}(a_t)$ as the *habitual* network, as it corresponds to a form of fast decision-making, quickly evaluating and selecting a action; in contrast with the more *deliberative* system that includes future imagination via MC tree traversals [29].

### 3.3 State precision and top-down attention

One of the key elements of our framework is the state transition model $P_{\theta_s}(s_t|s_{t-1}, a_{t-1})$, that belongs to the agent's generative model. In our implementation, we take $s_t \sim \mathcal{N}(\mu, \sigma^2/\omega_t)$, where the multidimensional $\mu$ and $\sigma$ come from the linear and softplus units (respectively) of a neural network with parameters $\theta_s$ applied to $s_{t-1}$, and, importantly, $\omega_t$ is a scalar *precision factor* (c.f. Fig. 1A) modulating the uncertainty on the agent's estimate of the hidden state of the environment [8]. We model the precision factor as a simple logistic function of the belief update about the agent's current policy,

$$\omega_t = \frac{\alpha}{1 + e^{-\frac{b - D_{t-1}}{c}}} + d \,, \tag{10}$$

where $D_t = D_{\text{KL}}[Q_{\phi_a}(a_t) \parallel P(a_t)]$ and $\{\alpha, b, c, d\}$ are fixed hyper-parameters. Note that $\omega_t$ is a monotonically decreasing function of $D_{t-1}$, such that when the posterior belief about the current policy is similar to the prior, precision is high.

In cognitive terms, $\omega_t$ can be thought of as a means of *top-down attention* [30], that regulates which transitions should be learnt in detail and which can be learnt less precisely. This attention mechanism acts as a form of resource allocation: if $D_{\text{KL}}[Q_{\phi_a}(a_t) \parallel P(a_t)]$ is high, then a habit has not yet been formed, reflecting a generic lack of knowledge. Therefore, the precision of the prior $P_{\theta_s}(s_t|s_{t-1}, a_{t-1})$ (i.e., the belief about the current state before a new observation $o_t$ has been received) is low, and less effort is spent learning $Q_{\phi_s}(s_t)$.

In practice, the effect of $\omega_t$ is to *incentivize disentanglement* in the latent state representation $s_t$ – the precision factor $\omega_t$ is somewhat analogous to the $\beta$ parameter in $\beta$-VAE [31], effectively pushing the state encoder $Q_{\phi_s}(s_t)$ to have independent dimensions (since $P_{\theta_s}(s_t|s_{t-1}, a_{t-1})$ has a diagonal covariance matrix).[2] As training progresses and the habitual network becomes a better approximation of $P(a_t)$, $\omega_t$ is gradually increased, implementing a natural form of precision annealing.

## 4 Results

First, we present the two environments that were used to validate our agent's performance.

**Dynamic dSprites** We defined a simple 2D environment based on the dSprites dataset [32, 31]. This was used to $i$) quantify the agent's behavior against ground truth state-spaces and $ii$) evaluate the agent's ability to disentangle state representations. This is feasible as the dSprites data is designed for characterizing disentanglement, using a set of interpretable, independent ground-truth latent factors. In this task, which we call *object sorting*, the agent controls the position of the object via 4 different actions (right, left, up or down) and is required to sort single objects based on their shape (a latent factor). The agent receives reward when it moves the object across the bottom border, and the reward value depends on the shape and location as depicted in Fig. 2A. For the results presented in Section 4, the agent was trained in an on-policy fashion, with a batch size of 100.

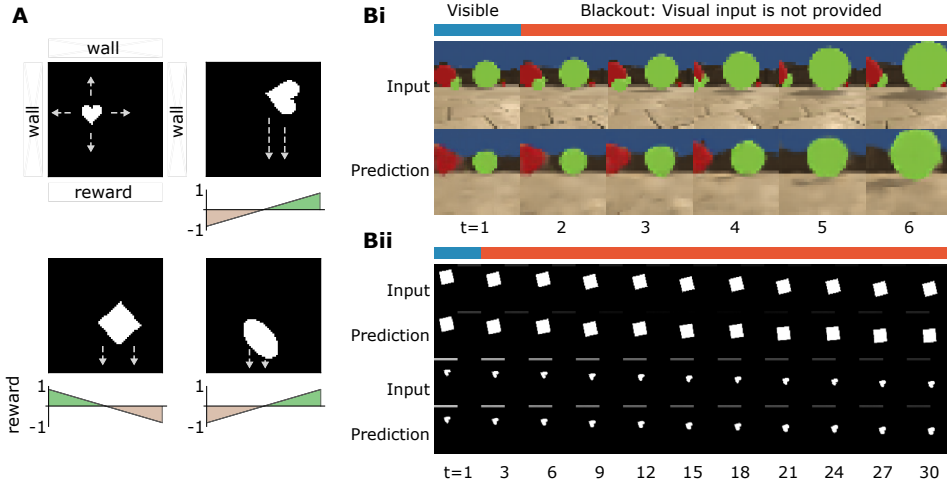

Figure 2: **A:** The proposed *object sorting* task based on the dSprites dataset. The agent can perform 4 actions; changing the position of the object in both axis. Reward is received if an object crosses the bottom boarder and differs for the 3 object shapes. **B:** Prediction of the visual observations under motion if input is hidden in both (**i**) AnimalAI and (**ii**) dynamic dSprites environments.

**Animal-AI**    We used a variation of *'preferences'* task from the Animal-AI environment [33]. The complexity of this, partially observable, 3D environment is the ideal test-bed for showcasing the agent's reward-directed exploration of the environment, whilst avoiding negative reward or getting stuck in corners. In addition, to test the agent's ability to rely on its internal model, we used a *'lights-off'* variant of this task, with temporary suspension of visual input at any given time-step with probability $R$. For the results presented in Section 4, the agent was trained in an off-policy fashion due to computational constraints. The training data for this was created using a simple rule: move in the direction of the greenest pixels.

In the experiments that follow, we encode the actual reward from both environments as the prior distribution of future expected observations $\log P(o_\tau|\pi)$ or, in active inference terms, the expected outcomes. This is appropriate because the active inference formulation does not differentiate reward from other types of observations, but it rather defines certain (future) observations (e.g. green color in Animal-AI) as more desirable given a task. Therefore, in practice, rewards can be encoded as observations with higher prior probability using $\log P(o_\tau|\pi)$.

We optimized the networks using ADAM [34], with loss given in Eq. (4) and an extra regularization term $D_{\mathrm{KL}}\big[Q_{\phi_s}(s_t) \parallel N(0,1)\big]$. The explicit training procedure is detailed in the supplementary material. The complete source-code, data, and pre-trained agents, is available on GitHub (https://github.com/zfountas/deep-active-inference-mc).

## 4.1   Learning environment dynamics and task performance

We initially show – through a simple visual demonstration (Fig. 2B) – that agents learn the environment dynamics with or without consistent visual input for both dynamic dSprites and AnimalAI. This is further investigated, for the dynamic dSprites, by evaluating task performance (Fig. 3A-C), as well as reconstruction loss for both predicted visual input and reward (Fig. 3D-E) during training.

To explore the effect of using different EFE functionals on behavior, we trained and compared active inference agents under three different formulations, all of which used the implicit reward function $\log P(o_\tau)$, against a baseline reward-maximizing agent. These include $i$) beliefs about the latent states (i.e., terms a,b from Eq. 7), $ii$) beliefs about both the latent states and model parameters (i.e., complete Eq. 7) and $iii$) beliefs about the latent states, with a down-weighted reward signal. We found that, although all agents exhibit similar performance in collecting rewards (Fig. 3B), active inference agents have a clear tendency to explore the environment (Fig. 3C). Interestingly, our results also demonstrate that all three formulations are better at reconstructing the expected reward, in comparison to a reward-maximizing baseline (Fig. 3D). Additionally, our agents are capable of reconstructing the

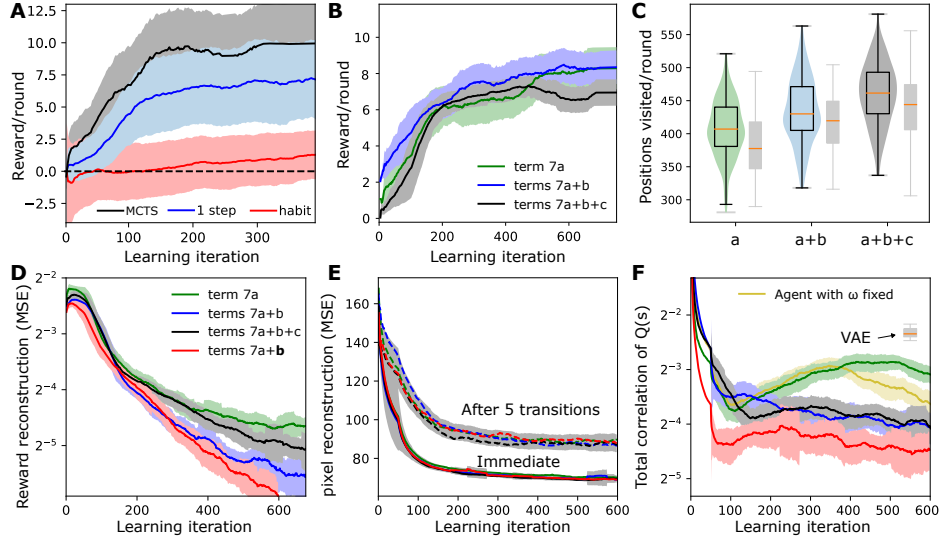

Figure 3: Agent's performance during on-policy training in the *object sorting* task. **A:** Comparison of different action selection strategies for the agent driven by the full Eq. (8). **B-C:** Comparison of agents driven by different functionals, limited to state estimations of a single step into the future. In C, the violin plots represent behavior driven by $P(a_t)$ (the planner) and the gray box plots driven by the habitual network $Q_{\phi_a}(a_t)$. **D-F:** Reconstruction loss and total correlation during learning for 4 different functionals. In A, B and D-F, the shaded areas represent the standard deviation.

current observation, as well as predicting 5 time-steps into the future, for all formulations of EFE, with similar loss with the baseline (Fig. 3E).

## 4.2 Disentanglement and transition learning

Disentanglement of latent spaces leads to lower dimensional temporal dynamics that are easier to predict [35]. Thus, generating a disentangled latent space $s$ can be beneficial for learning the parameters of the transition function $P_{\theta_s}(s_{t+1}|s_t, a_t)$. Due to the similarity between the precision term $\omega_t$ and the hyper-parameter $\beta$ in $\beta$-variational autoencoders (VAEs) [31] discussed in Sec. 3.3, we hypothesized that $\omega_t$ could play an important role in regulating transition learning. To explore this hypothesis, we compared the total correlation (as a metric for disentanglement [36]) of latent state beliefs between $i$) agents that have been trained with the different EFE functionals, $ii$) the baseline (reward-maximizing) agent, $iii$) an agent trained without top-down attention (although the average value of $\omega_t$ was maintained), as well as $iv$) a simple VAE that received the same visual inputs. As seen in Fig. 3F, all active inference agents using $\omega_t$ generated structures with significantly more disentanglement (see traversals in supp. material). Indeed, the performance ranking here is the same as in Fig. 3D, pointing to disentanglement as a possible reason for the performance difference in predicting rewards.

## 4.3 Navigation and planning in reward-based tasks

The training process in the dynamic dSprites environment revealed two types of behavior. Initially, we see epistemic exploration (i.e., curiosity), that is overtaken by reward seeking (i.e., goal-directed behavior) once the agent is reasonably confident about the environment. An example of this can be seen in the left trajectory plot in Fig. 4Ai, where the untrained agent – with no concept of reward – deliberates between multiple options and chooses the path that enables it to quickly move to the next round. The same agent, after 700 learning iterations, can now optimally plan where to move the current object, in order to maximize potential reward, $\log P(o_\tau|\pi)$. We next investigated the sensitivity when deciding, by changing the threshold $T_{dec}$. We see that changing the threshold has clear implications for the distribution of explored alternative trajectories i.e., number of simulated states (Fig. 4Aii). This plays an important role in the performance, with maximum performance found at $T_{dec} \approx 0.8$ (Fig. 4Aiii).

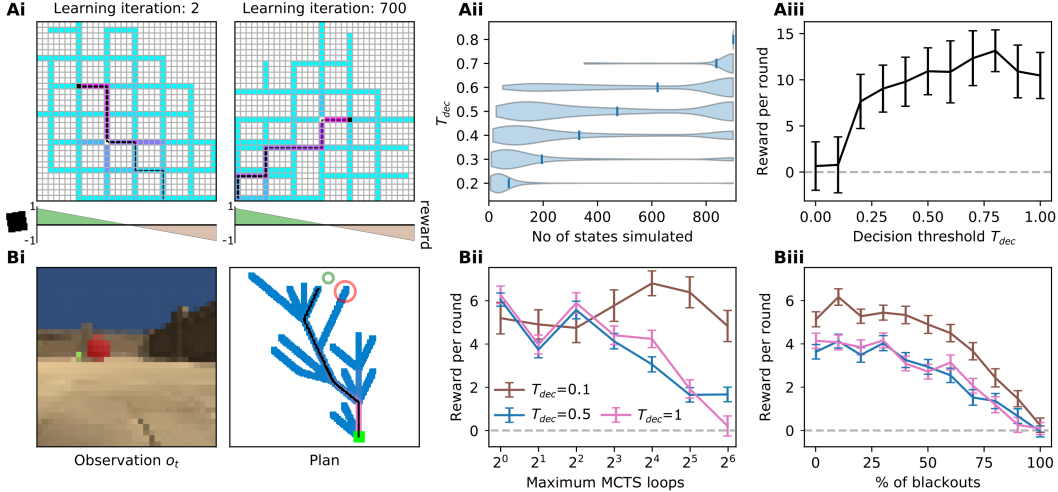

Figure 4: Agent's planning performance. **A:** Dynamic dSprites. **i)** Example planned trajectory plots with number of visits per state (blue-pink color map) and the selected policy (black lines). **ii)** The effect of decision threshold $T_{\text{dec}}$ on the number of simulated states and **iii)** the agent's performance. **B:** Animal-AI. **i)** Same as in A. **ii)** System performance over hyper-parameters and **iii)** in the *lights-off* task. Error bars in Aiii denote standard deviation and in B standard error of the mean.

Agents trained in the Animal-AI environment also exhibit interesting (and intelligent) behavior. Here, the agent is able to make complex plans, by avoiding obstacles with negative reward and approaching expected outcomes (red and green objects respectively, Fig. 4Bi). Maximum performance can be found for 16 MCTS loops and $T_{dec} \approx 0.1$ (Fig. 4Bii; details in the supplementary material). When deployed in *lights-off* experiments, the agent can successfully maintain an accurate representation of the world state and simulate future plans despite temporary suspension of visual input (Fig. 2B). This is particularly interesting because $P_{\theta_s}(s_{t+1}|s_t, a_t)$ is defined as a feed-forward network, without the ability to maintain memory of states before $t$. As expected, the agent's ability to operate in this set-up becomes progressively worse the longer the visual input is removed, while shorter decision thresholds are found to preserve performance longer (Fig. 4Biii).

## 4.4 Comparison with model-free reinforcement learning agents

To assess the performance of the active inference agent with respect to baseline (model-free) RL agents, we employed OpenAI's `baselines` [37] repository to train DQN [38], A2C [39] and PPO2 [40] agents on the Animal-AI environment. The resulting comparison is shown in Fig. 5. Our experiments highlight that, given the same number of training episodes (2M learning iterations), the active inference agent performs considerably better than DQN and A2C, and is comparable to PPO2 (all baselines were trained with default settings). In addition, note that of the two best-performing agents (DAIMC and PPO2), DAIMC has substantially less variance across training runs, indicating a more stable learning process. Nonetheless, these comparisons should be treated as a way of illustrating the applicability of the active inference agent operating in complex environments, and not as a thorough benchmark of performance gains against state-of-the-art RL agents.

## 5 Concluding Remarks

The attractiveness of active inference inherits from the biological plausibility of the framework [4, 41, 42]. Accordingly, we focused on scaling-up active inference inspired by neurobiological structure and function that supports intelligence. This is reflected in the hierarchical generative model, where the higher-level policy network contextualizes lower-level state representations. This speaks to a separation of temporal scales afforded by cortical hierarchies in the brain and provides a flexible framework to develop biologically-inspired intelligent agents.

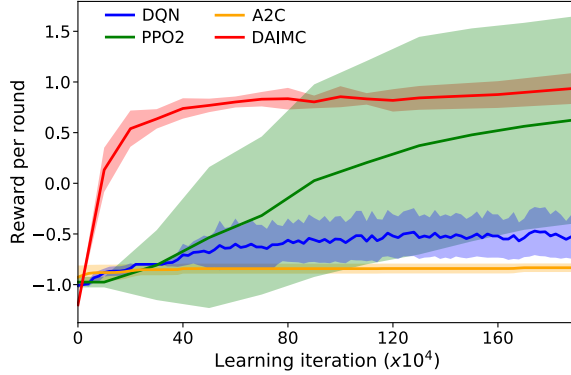

Figure 5: Comparison of the performance of our agent (DAIMC) with DQN, A2C and PPO2. The bold line represents the mean, and shaded areas the standard deviation over multiple training runs.

We introduced MCTS for tackling planning problems with vast search spaces [23, 43, 24, 44, 45]. This approach builds upon Çatal *et al.*'s [46] deep active inference proposal, to use tree search to recursively re-evaluate EFE for each policy, but is computationally more efficient. Additionally, using MCTS offers an *Occam's window* for policy pruning; that is, we stop evaluating a policy path if its EFE becomes much higher than a particular upper confidence bound. This pruning drastically reduces the number of paths one has to evaluate. It is also consistent with biological planning, where agents adopt brute force exploration of possible paths in a decision tree, up to a resource-limited finite depth [47]. This could be due to imprecise evidence about different future trajectories [48] where environmental constraints subvert evaluation accuracy [49, 50] or alleviate computational load [51]. Previous work addressing the depth of possible future trajectories in human subjects under changing conditions shows that both increased cognitive load [50] and time constraints [52, 53, 49] reduce search depth. Huys *et al.* [51] highlighted that in tasks involving alleviated computational load, subjects might evaluate only subsets of decision trees. This is consistent with our experiments as the agent selects to evaluate only particular trajectories based on their prior probability to occur.

We have shown that the precision factor, $\omega_t$, can be used to incorporate uncertainty over the prior and enhances disentanglement by encouraging statistical independence between features [54–57]. This is precisely why it has been associated with attention [58]; a signal that shapes uncertainty [59]. Attention enables flexible modulation of neural activity that allows behaviorally relevant sensory data to be processed more efficiently [60, 61, 30]. The neural realizations of this have been linked with neuromodulatory systems, e.g., cholinergic and noradrenergic [62–66]. In active inference, they have been associated specifically with noradrenaline for modulating uncertainty about state transitions [8], noradrenergic modulation of visual attention [67] and dopamine for policy selection [4, 67].

An important piece of future work is to more thoroughly compare the performance of DAIMC agent to reward-maximizing agents. That is, if the specific goal is to maximize reward, then it is not clear whether deep active inference (i.e., full specification of EFE) has significant performance benefits over simpler reward-seeking agents (i.e., using only Eq. 7a) or other model-based RL agents [68, 69] (c.f. Sec. 4.4). We emphasize, however, that the primary purpose of the active inference framework is to serve as a model for biological cognition, and not as an optimal solution for reward-based tasks. Therefore, we have deliberately not focused on benchmarking performance gains against state-of-the-art RL agents, although we hypothesize that insights from active inference could prove useful in complex environments where either reward maximization isn't the objective, or in instances where direct reward maximization leads to sub-optimal performance.

There are several extensions that can be explored, such as testing whether performance would increase with more complex, larger neural networks, e.g., using LSTMs to model state transitions. One could also assess if including episodic memory would finesse EFE evaluation over a longer time horizon, without increasing computational complexity. Future work should also test how performance shifts if the objective of the task changes. Lastly, it might be neurobiologically interesting to see whether the generated disentangled latent structures are apt for understanding functional segregation in the brain.

## 6 Broader impact

Our deep active inference agent – equipped with MC methods – provides a flexible framework that may help gain new insights into the brain by simulating realistic, biologically-inspired intelligent agents. General contributions of this framework include helping bridge the gap between cognitive science and deep learning and providing an architecture that would allow psychologists to run more realistic experiments probing human behavior. Specifically, we hope that simulating this agent will allow us use the neural network gradients to make predictions about the underlying physiology associated with behaviors of interest and formulate appropriate hypothesis. We believe this architecture may also help elucidate complex structure-function relationships in cognitive systems through manipulation of priors (under the complete class theorem). This would make it a viable (scaled-up) framework for understanding how brain damage (introduced in the generative model by changing the priors) can affect cognitive function, previously explored in discrete-state formulations of active inference [67, 70].

A potential (future) drawback is that this model could be used to exploit people's inherent cognitive biases, and as such could potentially be used by bad actors trying to model (and then profit from) human behavior.

## 7 Acknowledgements

The authors would like to thank Sultan Kenjeyev for his valuable contributions and comments on early versions of the model presented in the current manuscript and Emotech team for the great support throughout the project. NS was funded by the Medical Research Council (MR/S502522/1). PM and KJF were funded by the Wellcome Trust (Ref: 210920/Z/18/Z - PM; Ref: 088130/Z/09/Z - KJF).

## Footnotes

[2]In essence, for the parameter ranges of interest $\omega_t$ induces a near-linear monotonic increase in $D_{\text{KL}}$, akin to the linear increase induced by $\beta$ in $\beta$-VAE.

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
