[Supplementary Material]

# 8 Supplementary material

## 8.1 Expected free energy derivation

Here, we provide the steps needed to derive Eq. (8) from Eq. (7). The term (7b) can be re-written as:

$$\mathbb{E}_{\tilde{Q}}\left[\log Q(s_\tau|\pi) - \log Q(s_\tau|o_\tau,\pi)\right] =$$
$$= \mathbb{E}_{Q(\theta|\pi)Q(s_\tau|\theta,\pi)Q(o_\tau|s_\tau,\theta,\pi)}\left[\log Q(s_\tau|\pi) - \log Q(s_\tau|o_\tau,\pi)\right]$$
$$= \mathbb{E}_{Q(\theta|\pi)}\left[\mathbb{E}_{Q(s_\tau|\theta,\pi)}\log Q(s_\tau|\pi) - \mathbb{E}_{Q(s_\tau,o_\tau|\theta,\pi)}\log Q(s_\tau|o_\tau,\pi)\right]$$
$$= \mathbb{E}_{Q(\theta|\pi)}\left[\mathbb{E}_{Q(o_\tau|\theta,\pi)}H(s_\tau|o_\tau,\pi) - H(s_\tau|\pi)\right],$$

where we have only used the definition $\tilde{Q} = Q(o_\tau, s_\tau, \theta|\pi)$, and the definition of the standard (and conditional) Shannon entropy.

Next, the term (7c) can be re-written as:

$$\mathbb{E}_{\tilde{Q}}\left[\log Q(\theta|s_\tau,\pi) - \log Q(\theta|s_\tau,o_\tau,\pi)\right] =$$
$$= \mathbb{E}_{Q(s_\tau,\theta,o_\tau|\pi)}\left[\log Q(o_\tau|s_\tau,\pi) - \log Q(o_\tau|s_\tau,\theta,\pi)\right]$$
$$= \mathbb{E}_{Q(s_\tau|\pi)Q(o_\tau|s_\tau,\pi)}\log Q(o_\tau|s_\tau,\pi)$$
$$\quad - \mathbb{E}_{Q(\theta|\pi)Q(s_\tau|\theta,\pi)Q(o_\tau|s_\tau,\theta\pi)}\log Q(o_\tau|s_\tau,\theta,\pi)$$
$$= \mathbb{E}_{Q(\theta|\pi)Q(s_\tau|\theta,\pi)}H(o_\tau|s_\tau,\theta,\pi) - \mathbb{E}_{Q(s_\tau|\pi)}H(o_\tau|s_\tau,\pi),$$

where the first equality is obtained via a normal Bayes inversion, and the second via the factorization of $\tilde{Q}$. These two terms can be directly combined to obtain Eq. (8). With this expression at hand, the only problem that remains is estimating these quantities from the outputs of all neural networks involved. We provide some information here, in addition to that in Sec. 3.1.

For Eq. (8b), $H(s_\tau\,|\,\pi)$ is estimated sampling from the transition network, and $H(s_\tau\,|\,o_\tau,\pi)$ from the encoder network (both parameterised with Gaussians, so entropies can be calculated from log-variances). For the first term in Eq. (8c) we sample several $\theta$ from the MC-dropouts and several $s_\tau$ from the transition network; then average the entropies $H(o_\tau\,|\,s_\tau,\theta,\pi)$ (which are closed-form since $o_\tau$ is Bernoulli-distributed) over the $(\theta, s_\tau)$ samples. For the second term, we fix the $\theta$ and sample multiple $s_\tau$ (so that, effectively, $p(o|s) = \sum_\theta p(o|s,\theta)p(\theta)$ is approximated with a single MC sample) and repeat the procedure. Although noisy, this estimator was found to be fast and suitable for training. Finally, note that in both cases the quantities computed correspond to a difference between the entropy of the average and the average of the entropies – which is the mutual information, a known part of the EFE.

## 8.2 Glossary of terms and notation

| Notation | Definition |
|---|---|
| $S$ | Set of all possible hidden states |
| $s_t$ | Hidden state at time $t$, random variable over $S$ |
| $s_{1:t}$ | Sequence of hidden states, $s_1, .., s_t$, random variable over $S^t$ |
| $O$ | Set of all possible observations |
| $o_t$ | Observation at time $t$, random variable over $O$ |
| $o_{1:t}$ | Sequence of observations, $o_1, .., o_t$, random variable over $O^t$ |
| $T$ | Number of time steps in episode, positive integer |
| $U$ | Set of all possible actions |
| $a_t$ | Action at time $t$, random variable over $U$ |
| $\Pi$ | Set of all allowable policies; i.e., sequences of actions, subset of $U^t$ |
| $\pi$ | Policy as defined by $(a_1, a_2, ..., a_T)$, random variable over $\Pi$ |
| $P_{\theta_s}(s_t\|s_{t-1}, a_{t-1})$ | Transition function; parameterized by $\theta_s$ |
| $P_{\theta_o}(o_t\|s_t)$ | Likelihood/observation function; parameterized by $\theta_o$ |
| $P_{\theta_{o,s}}(s_{1:T}, a_{1:T-1}, o_{1:T})$ | Generative model; factorized form $P(a_1)P(s_1) \prod_{t=2}^{T} P_{\theta_s}(s_t\|s_{t-1}, a_{t-1}) \prod_{t=1}^{T} P_{\theta_o}(o_t\|s_t)$ |
| $Q_{\phi_a}(a_t)$ | Approximate posterior over actions; parameterized by $\phi_a$. Dependency on $s_t$ has been dropped following standard variational inference notation. |
| $Q_{\phi_s}(s_t)$ | Approximate posterior over hidden states; parameterized by $\phi_s$. Dependency on $o_t$ has been dropped following standard variational inference notation. |
| $Q_{\phi}(s_{1:T}, a_{1:T-1})$ | Approximate posterior over actions and hidden states with mean-field assumptions; $\prod_{t=1}^{T} Q_{\phi_s}(s_t)Q_{\phi_a}(a_t)$ |
| $-\log P_{\theta}(o_t)$ | Negative log-likelihood; surprisal at time $t$ |
| $\mathbb{E}_{Q_{\phi}(s_t, a_t)} \left[ \log Q_{\phi}(s_t, a_t) - \log P_{\theta}(o_t, s_t, a_t) \right]$ | Variational free energy or evidence lower bound at time $t$. This can be decomposed to Eq. (5) using the appropriate factorization. |
| $Q_{\phi}(s_{1:T}, a_{1:T-1}, o_{1:T})$ | Approximate posterior with mean-field assumptions; $\prod_{t=1}^{T} Q_{\phi_s}(s_t)Q_{\phi_a}(a_t)P_{\theta_o}(o_t\|s_t)$ |
| $\mathbb{E}_{P(o_\tau\|s_\tau, \theta)}\mathbb{E}_{Q_{\phi}(s_\tau, \theta\|\pi)} \left[ \log Q_{\phi}(s_\tau, \theta\|\pi) - \log P(o_\tau, s_\tau, \theta\|\pi) \right]$ | Expected free energy, defined on $\Pi$, for some future time–point $\tau$. This is derived by taking an additional expectation $P(o_\tau\|s_\tau, \theta)$ where $\theta$ denotes random variable over learnt distribution $P_{\theta}(.\|\pi)$ |
| $\sigma$ | Softmax function or normalized exponential |
| $P(\pi)$ | Posterior distribution about policies via softmax function of the summed (negative) expected free energy over time; $\sigma\left( -\sum_{\tau>t} G(\pi, \tau) \right)$ |
| $P(a_t)$ | Posterior distribution about actions via summed probability of all policies that begin with a particular action, $a_t$ |

Table 1: Glossary of terms and notation

## 8.3 Training Procedure

The model presented here was implemented in Python and the library TensorFlow 2.0. We initialized 3 different ADAM optimizers, which we used in parallel, to allow learning parameters with different rates. The networks $Q_{\phi_s}, P_{\theta_o}$ were optimized using an initial learning rate of $10^{-3}$ and, as a loss function, the first two terms of Eq. (4). In experiments where regularization was used, the loss function used by this optimizer was adjusted to

$$
\begin{aligned}
L_{\phi_s,\theta_o} = &-\mathbb{E}_{Q(s_t)}\big[\log P(o_t|s_t;\theta_o)\big] + \gamma D_{\mathrm{KL}}\big[Q_{\phi_s}(s_t) \parallel P(s_t|s_{t-1}, a_{t-1};\theta_s)\big] \\
&+ (1-\gamma)D_{\mathrm{KL}}\big[Q_{\phi_s}(s_t) \parallel N(0,1)\big]\,,
\end{aligned}
\tag{11}
$$

where $\gamma$ is a hyper parameter, starting with value 0 and gradually increasing to 0.8. In our experiments, we found that the effect of regularization is only to improve the speed of convergence and not the behavior of the agent and, thus, it can be safely omitted.

The parameters of the network $P_{\theta_s}$ were optimized using a rate of $10^{-4}$ and only the second term of Eq. (4) as a loss. Finally, the parameters of $Q_{\phi_a}$ were optimized with a learning rate of $10^{-4}$ and only the final term of Eq. (4) as a loss. For all presented experiments and learning curves, batch size was set to 50. A learning iteration is defined as 1000 optimization steps with new data generated from the corresponding environment.

In order to learn to plan further into the future, the agents were trained to map transitions every 5 simulation time-steps in dynamic dSprites and 3 simulation time-steps in Animal-AI. Finally, the runtime of the results presented here is as follows. For the agents in the dynamic dSprites environment, training of the final version of the agents took approximately 26 hours per version (on-policy, 700 learning iterations) using an NVIDIA Titan RTX GPU. Producing the learning and performance curves in Fig. 3, took 10 hours per agent when the 1-step and habitual strategies were employed and approximately 4 days when the full MCTS planner was used (Fig. 3A). For the Animal-AI environment, off-policy training took approximately 9 hours per agent, on-policy training took 8 days and, the results presented in Fig. 4 took approximately 4 days, using an NVIDIA GeForce GTX 1660 super GPU (CPU: i7-4790k, RAM: 16GB DDR3).

## 8.4 Training algorithm

The following algorithm is described for a single environment ($batch = 1$), to maintain notation consistency with the main text, but can also be applied when $batch > 1$. This algorithm is exactly the same for both Dynamic dSprites and Animal-AI environments. Finally, for either off-policy or off-line training, the action applied to the environment (line 9) is drawn from a different policy or loaded from a pre-recorded data-set respectively.

---
**Algorithm 1** DAIMC on-policy training

---
1: **for** $t = 1, 2, \ldots,$ max iterations **do**
2:     Randomize environment and sample a new observation $\tilde{o}_t$.
3:     Run planner and compute prior policy $P(a_t)$.
4:     Compute $Q_{\phi_s}(s_t)$ using $\tilde{o}_t$.
5:     Compute $Q_{\phi_a}(a_t)$ using a sampled state $\tilde{s}_t \sim Q_{\phi_s}(s_t)$.
6:     Compute $D_t = D_{\mathrm{KL}}\big[Q_{\phi_a}(a_t) \parallel P(a_t)\big]$.
7:     Apply a gradient step on $\phi_a$ using $D_t$ as loss.
8:     Compute $\omega_{t+1}$ from Eq. (10) using $D_t$.
9:     Apply action $\tilde{a}_t \sim P(a_t)$ to the environment and sample a new observation $\tilde{o}_{t+1}$.
10:     Compute $\mu, \sigma$ from $P_{\theta_s}(s_{t+1}|\tilde{s}_t, \tilde{a}_t)$.
11:     Compute $Q_{\phi_s}(s_{t+1})$ using $\tilde{o}_{t+1}$.
12:     Apply a gradient step on $\theta_s$ using $D_{\mathrm{KL}}\big[Q_{\phi_s}(s_{t+1}) \parallel \mathcal{N}(\mu, \sigma^2/\omega_t)\big]$.
13:     Apply a gradient step on $\phi_s$, $\theta_o$ using $-\mathbb{E}_{Q(s_{t+1})}\big[\log P_{\theta_o}(o_{t+1}|s_{t+1})\big] + D_{\mathrm{KL}}\big[Q_{\phi_s}(s_{t+1}) \parallel \mathcal{N}(\tilde{\mu}, \tilde{\sigma}^2/\omega_t)\big]$.
14: **end for**

---

## 8.5 Model parameters

In both simulated environments, the network structure used was almost identical, consisting of convolutional, deconvolutional, fully-connected and dropout layers (Fig. 6). In both cases, the dimensionality of the latent space $s$ was 10. For the top-down attention mechanism, the parameters used were $\alpha = 2, b = 0.5, c = 0.1$ and $d = 5$ for the Animal-AI environment and $\alpha = 1, b = 25, c = 5$ and $d = 1.5$ for dynamic dSprites. The action space was $|\mathcal{A}| = 3$ for Animal-AI and $|\mathcal{A}| = 4$ for dynamic dSprites. Finally, with respect to the planner, we set $c_{\text{explore}} = 1$ in both cases, $T_{dec} = 0.8$ (when another value is not specifically mentioned), the depth of MCTS simulation rollouts was set to 3, while the maximum number of MCTS loops was set to 300 for dynamic dSprites and 100 for Animal-AI.

**Q(s):**
input: 64x64x1

- ReLU | size: 31x31x32, kernel: 3, strides: (2, 2) [Conv2D]
- ReLU | size: 15x15x32, kernel: 3, strides: (2, 2) [Conv2D]
- ReLU | size: 31x31x32, kernel: 3, strides: (2, 2) [Conv2D]
- ReLU | size: 7x7x64, kernel: 3, strides: (2, 2) [Conv2D]
- ReLU | size: 3x3x64, kernel: 3, strides: (2, 2) [Conv2D]
- ReLU | size: 256 [Fully-connected]
- rate: 0.5 [Dropout]
- ReLU | size: 256 [Fully-connected]
- rate: 0.5 [Dropout]
- ReLU | size: 256 [Fully-connected]
- rate: 0.5 [Dropout]
- size: 10 + 10 [Fully-connected]

**P(o|s):**
input: 10

- ReLU | size: 256 [Fully-connected]
- rate: 0.5 [Dropout]
- ReLU | size: 256 [Fully-connected]
- rate: 0.5 [Dropout]
- ReLU | size: 256 [Fully-connected]
- rate: 0.5 [Dropout]
- ReLU | size: 256 [Fully-connected]
- rate: 0.5 [Dropout]
- ReLU | size: 16x16x64, kernel: 3, strides: (2, 2) [DeConv2D]
- ReLU | size: 32x32x64, kernel: 3, strides: (2, 2) [DeConv2D]
- ReLU | size: 64x64x32, kernel: 3, strides: (2, 2) [DeConv2D]
- size: 64x64x1, kernel: 3, strides: (2, 2) [DeConv2D]

**P(s|s,a):**
input: 10+4

- ReLU | size: 512 [Fully-connected]
- rate: 0.5 [Dropout]
- ReLU | size: 512 [Fully-connected]
- rate: 0.5 [Dropout]
- ReLU | size: 512 [Fully-connected]
- rate: 0.5 [Dropout]
- size: 10+10 [Fully-connected]

**Q(a|s):**
input: 10

- ReLU | size: 128 [Fully-connected]
- ReLU | size: 128 [Fully-connected]
- size: 4 [Fully-connected]

Legend:
- Conv2D
- DeConv2D
- Fully-connected
- Dropout

Figure 6: Neural network parameters used for the dynamic dSprites experiments. For the Animal-AI experiments, the only differences are: i) the input layer of the network used for $Q_{\phi_s}(s)$ and output layer for $P_{\theta_o}(o_t|s_t)$ have shape $(32, 32, 3)$, ii) the input layer of $P_{\theta_s}(s_{t+1}|s_t, a_t)$ has shape $(10 + 3)$ and iii) the output layer of $Q_{\phi_a}(a_t)$ has a shape of $(3)$, corresponding to the three actions forward, left and right.

## 8.6 Examples of agent plans

**A)** Learning iteration: 2

**B)** Learning iteration: 700

Figure 7: Examples of consecutive plans in the dynamic dSprites environment during a single experiment.

Figure 8: Examples of plans in Animal-AI environment. Examples were picked randomly.

## 8.7 Examples of traversals

Figure 9: Latent space traversals for the full active inference agent optimized in the dynamic dSprites environment. Histograms represent distribution of values for 1000 random observations. The graphs on the right column represent correlation between each dimension of $s$ and the 6 ground truth features of the environment. This includes the 5 features of the dSprites dataset and reward, encoded in the top pixels shown in $s_9$.