[Reviews · NeurIPS 2020]

Review 1

Summary and Contributions: Presents a model of active inference, a normative theory of learning, action and perception, which is at a greater scale relative to a number of active inference models in the literature. Applies it to two tasks, one looking to demonstrate disentanglement and one showing planning in complex environments. Main contributions relative to prior work: i) Inclusion of a habitual network (novel to active inference, not literature) ii) State transition precision iii) MC dropout for parameter distribution estimation (novel to active inference, not literature) This paper should be accpeted with modification

Strengths: The authors focus on all aspects of expected free energy, not done in previous literature and is key to understanding the effectiveness of functional in real-world tasks. The authors make some claim to address biological plausibility rather than just scaling, but this is questionable. The calculating of expected free energy is initially impressive and it seems as though MC dropout provides an efficient approach to calculating functionals made up of expectations and entropies, although I have some questions about the validity of this in the weaknesses section. The double Bayesian inversion of state information gain is also nice (reducing to entropy over observations), I had not seen this before. But why not just write it out in terms of entropies over states, and use same procedure? The links between precision in action and state transitions are fantastic - and the link the beta-VAE is compelling, although more detail on this would be warranted. The results show some of the expected behaviours - an effect of precision on disentanglement and a transition from exploration to exploitation.

Weaknesses: The main issue is the contribution. Scaling active inference has been done before, and while this may be an improvement on previous work, it does not make any fundamental progressions. If you had pushed the bio-angle, this may have worked, but this was put to the wayside. The results are also weak. The tasks are appropriate, show different aspects of model have intended effects (most of the time). But there are no comparisons to any other approach. It could well be the case that your algorithm takes 10^6 times longer than the most basic RL algorithm, and we would have no way of knowing. I do not think benchmarks are the metric around which we want to optimise our research, but they serve a purpose of providing context for the effectiveness of methods. There is no described relation between the normative goal of minimise expected surprise and minimising expected free energy - the relationship between these quantities seems unclear and the provided reference does not help clarify the matter. Could the authors state simply how expected surprise relates to expected free energy, and why the former entails the latter? The description of how you calculate 8b and 8c needs expanding. See additional comments, but I would like to see this addressed in response before I'd be happy with the paper being published. The use of UCB in MCTS is disappointing given you are working in an intrinsically Bayesian framework - these issues could have been addressed within the formalism. The combination of amortised action selection and MCTS is rather arbitrary and not particularly justified. It is also misleading to call it a habitual network as it never actually selects actions, just initialises certain aspects of the plan. No description of MC dropout as Bayes, and in general the training procedure could be described as an algorithm.

Correctness: Line 45 - 46, please elaborate or play down (or at least use references for) “is consistent with planning strategies employed by biological agents”. The use of the MC tree search is assumed to be consistent with strategies employed by biological agents.

Clarity: You define MC abbreviation in the abstract, but just because it is used so widely throughout, please defined when used in text first time (line 43). Description of active inference is very good for space used. Overall, a clearly written paper, besides the clarifications stated above.

Relation to Prior Work: The authors mostly do fair justice to prior work. Less explored is the relation to work on combing amortised action with planning. See for example [1,2,3] (although 3 came out after this was submitted). 1 Piché A, Thomas V, Ibrahim C, Bengio Y, Pal C. Probabilistic planning with sequential monte carlo methods. InInternational Conference on Learning Representations 2018 2) An Inference Perspective on Model-Based Reinforcement Learning Joseph Marino, Yisong Yue ICML 2019 Workshop on Generative Modeling and Model-Based Reasoning for Robotics and AI 3) Control as Hybrid Inference Alexander Tschantz, Beren Millidge, Anil K. Seth, Christopher L. Buckley ICML 2020 Workshop on Theoretical foundation of RL

Reproducibility: No

Additional Feedback: The description of how you calculate 8b and 8c needs expanding. It seems to me that you are sampling to generate multiple distributions, taking the entropy of these, and then averaging the entropy. Further discussion needs to be made as to how this is the same as taking the entropy of the average (as opposed to the average of the entropy) Moreover, if this is the case and I am mistaken, the ability to perform the expectations required in equation 8b and 8c and end up with valid Gaussian distribution is by no means trivial. The only way I can see it working is as some property of MC. Either way, please describe how your averaging procedure leads to Gaussian distributions, as this is perhaps the most technical part of the paper. There is a substantial literature on approximating expected information, and I am aware of none which have gotten over this problem - hence resorting to entropy estimation procedures or writing out the expected KL in a different form. If this provides a valid procedure then it would be a contribution to the field in its own r


Review 2

Summary and Contributions: The authors present an active-inference approach to model-based control and learning inspired by recent developments in theoretical neuroscience. Their method is rooted in variational inference. To evaluate, they introduce a new agent environment, Dynamic d-Sprites, and also evaluate in Animal AI. As the authors admit, they do not aim to "compete" directly with agent models that solely maximize reward (such as most reinforcement learning agents), and so they do not compare rewards gained to a purely RL baseline in any task. However, I feel that this gives the novelty of their work its space to shine.

Strengths: Predictive performance, model learning, and task performance are all evaluated, a definite plus of the paper. If anything, the "common currency" of information-theoretic units for both reward and exploration, enabling active inference to "finesse" the exploration-exploitation dilemma, deserves more emphasis. Their method is rooted in variational inference, justifying their objective functions and optimization procedure. Agents trained via the authors' active inference procedure have predictive reconstruction capabilities (imputing missing frames of video) as well as acting rationally to seek rewards in the environment. The authors' paper is one of the two or three papers I have seen over the past several years that attempt to "scale up" active inference techniques beyond GridWorld tasks whose solutions can be computed quickly in closed form.

Weaknesses: As noted under Clarity and Comments, the paper perhaps suffers from the "curse of knowledge" in expecting that the reader to have read many active inference papers. In particular, constructions such as $\tilde{Q} = Q(o_\tau, s_\tau, \theta \mid \pi)$ should be expanded and defined explicitly (what is the likelihood being combined with the variational distribution on states?) rather than by reference to previous work. I hope space for such an equation can be found. Similarly, $P(o_\tau)$ should be explicitly defined and notated as a distribution over desired outcomes, rather than the marginal of the perceptual model's joint distribution. I don't see any further usage of the $\gamma$ parameter to the softmin in Equation 3. Did the authors simply hard-code its value for their experiments? I would have liked to see $\gamma$ treated using variational inference to control the planning horizon, as in earlier active-inference work. As the authors themselves admit, they do not really evaluate against "state of the art" reinforcement learning agents. Since I think their supplemental material and figures provide enough information to reproduce their work, I consider this only a slight drawback.

Correctness: No prior distribution was given for $P(\theta)$, and so I would ask the authors to explain how a posterior distribution can be formed without a random variable in Equation 7c. On line 168, I would think that *more* effort should be spent learning to map from observations to states when the prior is weak.

Clarity: The paper is readable and clearly written. However, if I may nitpick as a reviewer, it might be somewhat preferable in communicating with the standard NeurIPS audience to use much of the notation and terminology of variational inference, and put the connections to Friston's free-energy principle in an appendix. Mathematically, this changes nothing, but in literary and social terms, I want the community to be able to appreciate this paper without requiring a seminar in a recent branch of neuroscience. Encoder networks representing the approximate posterior distribution should, customarily, be written as conditional densities: $Q_{\phi_s}(s_t \mid o_t) is indicated in Figure 1a, whereas $Q_{\phi_s}(s_t)$ would usually denote a mean-field variational distribution. The final sentence in lines 183-184 should be moved to the top of the section, since it applies to both experiments. The paragraph from 251-256 is the neuroscience version of technobabble, but oh well, that seems to be a business we're all in these days. There are occasional missing articles. I would strongly prefer that the authors write in a mathematical notation suitable for most readers and authors at NeurIPS who work with variational inference methods, and so I am disappointed to hear they would prefer to stay in the "active inference ghetto" at potential detriment to the impact of their work. I am changing my score to reflect their response to my feedback.

Relation to Prior Work: The work is sufficiently novel that it cites and discusses most of the related prior work on the active-inference side of things. It could have done more to cite model-based RL and variational control work, such as https://arxiv.org/abs/1805.00909, https://arxiv.org/abs/1811.01132, and similar.

Reproducibility: Yes

Additional Feedback: Interesting future work would compare the degree of "reward shaping" required in active inference versus RL agents. Since ActInf and RL have different objective functions (exploration + exploitation + curiosity for the former, exploitation-only for the latter), experiments should be conducted in which the "ad hoc" components of RL are brought to the fore and compared to their more principled counterparts in Active Inference. Since the paper does not "compete" with RL, I would advise the authors to try to refine Dynamic dSprites, AnimalAI, or something similar into a "killer app" that showcases the usefulness of active inference. This is an interesting sub-field that deserves not to be quashed by benchmark-chasing tendencies.


Review 3

Summary and Contributions: This paper proposes a variational neural net architecture and training framework for learning agents which can mix a form of planning (MCTS) and habitual policy. The authors argue that a particular element of the architecture which dampens or not the learning on specific time steps helps to achieve better disentangled representations.

Strengths: This is a fairly complex system which combines many ingredients from the recent deep learning and RL literature. Getting inspiration from cognitive neuroscience to build learning agent is clearly a good thing which motivates some of the elements proposed.

Weaknesses: The complexity of the system makes it difficult to understand many details, and I often found myself wondering why specific choices were made. The main weakness is thus clarity. Another weakness is the experiments. It would have helped to pick at least some setup which has already been explored by others using recent deep learning / deep RL techniques to have a more reliable point of comparison. One reason why it was difficult for me to follow the first few sections is that they rely a lot on the first 19 cited papers which all come from the same group, and it was difficult for me to grasp the motivations and significance of this work without going into that literature. Clearly, the paper could be improved by trying to make it more self-contained and internally justified (although I understand this is challenging given the length constraints), but I'd rather have that and better understand the deep motivations and innovations and have more of the experiments described in appendices.

Correctness: There are several places where I was not sure to understand the math. For example, I wonder if several probabilities which are presented as unconditional should instead be viewed as conditional: the Q's should probably be conditional (respectively on the state or the observation) and I don't understand what it means to talk about an unconditional P(action), maybe it should also be conditioned on the past, somehow. Similarly, on the inlined eqn between eqn 6 and eqn 7, I see a \pi conditioning the LHS but not the RHS.

Clarity: Not for a reader like me. I have already mentioned several things above. Other places where I stumbled include the parts about justifying the EFE as an objective for picking policies (for example, always going to same old place with no surprise does not seem like a very good policy), and the line 113 also about rewards stating that P(o|pi) = P(o) which seems weird to me, and generally why log P(o) should be a good reward. It was not clear at all to me either why the scaling by omega would help to disentangle the representations. Also, calling omega scaling an attention mechanism seems a bit stretched. Usually, attention refers to focusing on one of many elements in the current input (or memory). Here omega scales the importance given to different time steps, so it is indeed a form of attention, but not what one expects with the phrase <top-down attention>.

Relation to Prior Work: Comparisons to other work on unsupervised disentangling or unsupervised RL would clearly help the paper. How does the proposed EFE compare with more standard methods used for exploration and intrinsic rewards in RL? Making such comparisons (or citing them if they already exist) would really help better justify the approach.

Reproducibility: No

Additional Feedback: Thanks for the rebuttal feedback. I still think that the paper's accessibility to mainstream NeurIPS readers could be improved and that actual comparisons with SOTA standard RL methods are needed, where the evaluation is done not by the authors of this paper but in published benchmarks.


Review 4

Summary and Contributions: The paper has scaled up the implementation of active inference, a recent and well-known hypothesis is Neuroscience, through deep networks and Monte Carlo methods. The implementation has been tested on two toy data-sets much larger than previously tested data-sets for active inference.

Strengths: The paper tackles a problem that is currently a subject of interest for many in the computational neuroscience community. It is clearly written especially compared to other papers with the subject of active inference (or at least ones that I have read). It is also quite honest about the claims and has acknowledged a few of its weak points. While the results are not complete (see weakness), but the test benchmarks are acceptable even with the standards of AI/ML. Finally, I think the main strength of the paper is its relevance to the NeurIPS community especially to those who are working at the intesrection of AI and Neurosceince.

Weaknesses: While the authors mentioned this in their final section, I still think the result of a reward maximizing agent should be included in the results to provide something to compare with. I don't think they should beat those agents, but it is really important to know the optimal (or good enough) reward rate. For example in fig 3a, it is important to know that whether a reward-seeking agent obtains 15 rewards/round or 100 rewards/round. Also, the number of trials that the agent needs to gain acceptable performance seems too high to me (700 in section 4.3). One of the hallmarks of biological intelligence is few-shot learning. I don't think it should reach the optimal solution very fast, but reaching a sub-optimal (or good enough) performance in a few trials is desirable.

Correctness: To my knowledge mainly yes. However, I believe some of the claims about how the brain works are taken for granted. As far as I know, many of these mechanisms such as mental stimulation or attention are still pretty much unknown.

Clarity: Yes, it is well written especially compared to other active inference papers that I have read. However, similar to many previous works in active inference, handling reward is presented vaguely.

Relation to Prior Work: Yes

Reproducibility: Yes

Additional Feedback: As I mentioned in the limitations, I think it is good to provide the result of a reward-seeking agent just for showing how well (or bad) active inference agent performs. Also, the number of trials that the agent needs for navigation tasks (700) seems high to me. It would be great if the authors provide some explanation/comparison to show 700 is a reasonable number. Also, I appreciate if the authors provide intuition for eq 3. Moreover, I am afraid the stochasticity of equation (3) has made the agent eventually learn the navigation task in 700 trials and the minimization of surprise part did not contribute to learning. After rebuttal: Based on discussion with other reviewers, it looks like core concepts of active inference should be explained more in the paper to be more accessible for the broad machine learning community. If the paper gets accepted, I think the authors could make some room for better explaining active inference by reducing some pure-neuroscience paragraphs (for example in discussion).

[Author Response · NeurIPS 2020]

1  We thank the reviewers and AC for their thoughtful comments and thorough review. For this response, we have identified
2  a few common themes that have been raised by several reviewers, and address them in turn. We begin by discussing a
3  few major issues we believe are key for the reviewers' evaluation of the main contribution of the paper:

1. Reviewers #1, #3, and #4 request a more thorough evaluation against standard RL and reward-seeking agents. A
   comparison against a reward-seeking, exploitation-only agent is provided in Fig. 3b, showing that it performs
   similarly to the full Active Inference (AIF) agent (but with less effective exploration, as expected). We also tested
   agents from OpenAI's baselines repository on the Animal-AI environment, and found that (given the same number
   of training episodes) our agent performs considerably better than DQN and A2C, and comparable to PPO (all
   baselines with default settings). We will include detailed comparisons in the camera-ready version of the paper.

2. Reviewer #1 urges us to describe our calculation of Eqs. 8b and 8c. For 8b, $H(s_\tau \,|\, \pi)$ is estimated sampling
   from the transition network, and $H(s_\tau \,|\, o_\tau, \pi)$ from the encoder network (both parameterised with Gaussians, so
   entropies can be calculated from log-variances). For the first term in 8c we sample several $\theta$ from the MC-dropouts
   and several $s_\tau$ from the transition network; then average the entropies $H(o_\tau \,|\, s_\tau, \theta, \pi)$ (which are closed-form
   since $o_\tau$ is Bernoulli-distributed) over the $(\theta, s_\tau)$ samples. For the second term, we fix the $\theta$ and sample multiple
   $s_\tau$ (so that, effectively, $p(o|s) = \sum_\theta p(o|s, \theta)p(\theta)$ is approximated with a single MC sample) and repeat the
   procedure. (We also tried sampling several $\theta$ and averaging the distributions over $o_\tau$, which is possible because $o_\tau$
   is Bernoulli-distributed. Although noisier, the estimator described in the paper was faster and more suitable for
   training.) We agree with the reviewer's statement that the entropy of the average is not the same as the average of
   the entropies – and the difference between the two is the mutual information, which is known to be part of the EFE.
   We will describe this calculation in detail in the appendix.

3. Reviewers #2, #3 and #4, raise concerns about the clarity of our exposition, which we group in 3 items:
   - Regarding $P(o_\tau)$: At all times it should be conditioned on $\pi$, i.e. $P(o_\tau|\pi)$. This should not have appeared in
     lines 97 and 113 and we will update accordingly.
   - Regarding $\log P(o_\tau)$ as reward: We would like to clarify that in AIF the reward is not differentiated from
     other types of observations. Certain (future) observations (e.g. green color in Animal-AI) are considered more
     desirable given a task, so in practice rewards can be encoded as observations with higher prior probability
     using $\log P(o_\tau)$. We will make this conceptual point explicit in the camera-ready version.
   - Regarding $\tilde{Q} = Q(o_\tau, s_\tau, \theta \mid \pi)$: The expansion is shown in Eq. 8a under the expectation, although we agree
     it could benefit from being presented separately. We will mention it explicitly in the camera-ready version.

4. Reviewer #1 claims "scaling active inference has been done before." We agree with the reviewer that prior work
   has been done on this topic, but our contribution represents a technical (and qualitative) improvement over previous
   approaches. This is achieved by: 1) estimating all summands of EFE (line 41) and, 2) for the first time successfully
   training AIF agents on full-fledged, complex environments with visual input, multiple actions, and sparse rewards.
   We believe this constitutes a substantial improvement over the state of the art in AIF applications.

5. Reviewers #1 and #3 claim we do not provide enough details to reproduce our results. We would like to remind
   the reviewers we have uploaded our code to a public repository, which will be linked in the camera-ready version
   of this paper (line 198). Additionally, for clarity we will include pseudo-code for the algorithm in the appendix.

In addition, we would like to address a few minor comments:

6. Reviewers #1, #2 and #3 have suggested additional references for amortised action with planning, disentanglement
   and model-based RL. We will add these to the discussion in the camera-ready version.

7. Reviewer #2 suggests we could change the notation to be more in line with the variational inference literature.
   Although we agree with the reviewer's aims, given the space constraints and how much we rely on the AIF
   literature, we believe it would make the exposition denser and the links with prior AIF literature harder to track.
   Nonetheless, to make the paper more accessible to non-neuroscientists, in the camera-ready version we will add
   glossary to the appendix describing in detail what each symbol and probability distribution represent.

8. Reviewer #4 argues 700 trials "seems high," given that "one of the hallmarks of biological intelligence is few-shot
   learning." We agree with the reviewer, but emphasise that our agent starts 'from scratch' (i.e. with randomly
   initialised networks) each run, while biological organisms are fantastically able to form good priors that generalise
   and transfer between tasks. The extension of AIF to transfer learning remains an exciting avenue for future work.

9. The reviewers have identified a few grammatical errors, like occasional missing articles, misplaced sentences, or
   acronyms (like 'MC') that should be defined more explicitly. Additionally, reviewer #2 has identified redundant
   hyper-parameterisation (i.e. $\gamma$ in Eq. 3). We will address all of these in the camera-ready version of the paper.

53  We thank again the reviewers and AC for their work, which we are sure will improve the next version of this paper. We
54  hope this response addresses the core issues raised during the review process.

[Meta-Review · NeurIPS 2020]

This submission present a new model for active inference, a theory that combines action and perception into a single objective in the form of a free energy. The proposed approach combines several innovations that have not previously been applied to active inference problems, including the inclusion of a habitual network, the use of MC dropout to predict parameter belief updates, and a top-down mechanism that modulates the precision over state transitions. The authors evaluate the proposed mode on a newly developed agent environment (Dynamic d-Sprites) and also evaluate in Animal AI. Reviewers had mixed opinions on this submission. Three reviewers, who were familiar with literature on active inference, were positively predisposed. These reviewers noted that the paper is (comparatively) clearly written, makes a reasonable set of technical contributions, whilst acknowledging limitations in of the approach. Main criticisms came from the fourth reviewer, who found the exposition difficult to follow without prior knowledge of the many Friston-group papers that are cited. More broadly, reviewers noted problems with notation, found experimental results somewhat limited, and noted that comparisons to Deep RL baselines would be warranted (even if improved performance on RL tasks is not the primary intended contribution of this paper). All reviewers engaged in discussion post response. A substantial component of the discussion focused on whether the experiments include comparisons to baseline RL methods. The reviewers noted that model-based MCTS with only 7a is a reasonable proxy for existing RL methods. The AC would suggest that the authors point this out more explicitly. Reviewers are happy to hear that the authors are planning to include comparisons to DQN, A2C & PPO in the camera ready. Note that the fourth reviewer adjusted their score post discussion. Based on the reviews and discussion, this submission is just about above the a bar for acceptance. That said, having attempted to understand the proposed work, the AC agrees with comments about clarity and notation. It is of course infeasible (and not expected) to summarize all relevant existing literature, but the authors should make some attempt at a self-contained exposition. More importantly, all notation that is introduced should be explicit, particularly where it comes to each of the densities in the objective. It is unclear how Q(s_t) and P(o_t, s_t ; θ) in Eq (1) relate to Q(s_τ, θ | π) and P(o_τ, s_τ, θ | π). Similarly, it is not clear how P and Q in F_t relate to P and Q in G. Certain distributions such as Q(o_τ, s_τ, θ | π), are never defined in terms of conditionals. Other distributions, such as the encoders, have inputs that should be made notationally explicit (e.g. write Q_φ(s_t | o_t) and Q_φ(a_t | s_t) and not Q_φ(s_t) and Q_φ(a_t)). Finally the authors mix and match notation of parameterizations (i.e. Q_φ(…) vs P(… ; θ)). In addition to addressing these notational ambiguities, the AC would suggest that the authors begin by defining distributions P and Q over full trajectories (s_1, o_1, a_1, …, s_T, o_T), then express a free energy F in terms of this P and Q, then decompose this energy into a sum over time points F_t, and then explain how the prior over actions is defined in terms of G (and make it explicit how the distributions P and Q in G relate to the ones in F). This should make the exposition much easier to follow.